# Rethinking Newborn Screening: A Case of GALM Deficiency

**DOI:** 10.3390/ijns11020025

**Published:** 2025-04-11

**Authors:** Eva M. M. Hoytema van Konijnenburg, Silvia Radenkovic, Klaas Koop, Hubertus C. M. T. Prinsen, Monique de Sain-van der Velden

**Affiliations:** 1Section of Metabolic Diseases, Wilhelmina Children’s Hospital, University Medical Center Utrecht, 3584 CX Utrecht, The Netherlands; k.koop@umcutrecht.nl; 2Section Metabolic Diagnostics, Department of Genetics, University Medical Centre Utrecht, 3584 CX Utrecht, The Netherlands; s.radenkovic@umcutrecht.nl (S.R.); b.prinsen@umcutrecht.nl (H.C.M.T.P.); m.g.desain@umcutrecht.nl (M.d.S.-v.d.V.)

**Keywords:** galactosemia, newborn screening, GALM deficiency

## Abstract

Galactosemia is a group of hereditary disorders of galactose metabolism. A new type of galactosemia was discovered, caused by a deficiency in galactose mutarotase (GALM), which catalyzes the epimerization between beta- and alpha-D-galactose. All GALM-deficient patients reported in the literature (n = 44) had abnormal newborn screening (NBS) results or did not receive NBS (n = 2). We present the first patient with GALM deficiency who had negative NBS in the Netherlands and was identified at age 1.5 years during broad metabolic screening because of her global developmental delay, nystagmus, and a history of jaundice. Biochemical evaluation showed a significantly increased excretion of galactose (13,167 mmol/mol creatinine, upper limit of normal (ULN) 326) and galactitol (427 mmol/mol creatinine, ULN 71). Whole exome sequencing showed homozygous variants in *GALM* (c.424G>A p.(Gly142Arg)). A galactose-restricted diet was started, resulting in biochemical normalization. We present a comprehensive review of GALM-deficient patients, NBS data, and treatment. Different designs of galactosemia screening may lead to overlooking patients with GALM deficiency. Although the effects of lactose-restricted diet are largely unknown, a diet might prevent cataract in some patients.

## 1. Introduction

Galactosemia is an inherited metabolic disorder of carbohydrate (galactose) metabolism. Galactosemia type I is caused by a deficiency of galactose-1-phosphate uridyltransferase (GALT) [1]. GALT catalyzes the interconversion of galactose 1-phosphate and uridine-diphosphate (UDP)-glucose to glucose 1-phosphate and UDP-galactose [2,3] (Figure 1). GALT deficiency is further divided into the classical form, which has (almost) no detectable GALT enzyme activity, and a more frequently observed variant form (Duarte galactosemia), which has residual GALT enzymatic activity [4]. Apart from classic/Duarte galactosemia caused by GALT deficiency, deficiencies in other enzymes of the ‘Leloir pathway’ and galactose metabolism are known to cause galactosemia as well: type II galactosemia, caused by deficiency of galactokinase (GALK1), which phosphorylates galactose to galactose-1-phosphate deficiency [5], and type III galactosemia, caused by deficiency of UDP-galactose epimerase (GALE), which catalyzes the isomerization of UDP-galactose [6]. All genetic disorders in the Leloir pathway result in high galactose levels, which is then converted to galactitol by aldose reductase (Figure 1). Therefore, apart from increased galactose levels, galactosemia patients usually have increased galactitol.

More recently, a fourth type of galactosemia was discovered [7], caused by a deficiency in galactose mutarotase (GALM), whose major role is the epimerization between beta- and alpha-D-galactose (Figure 1). Although the GALM enzyme has been known for a long time [8], no relation with human disease was described before 2019 [7]. In the first report, eight patients with unexplained congenital galactosemia detected in newborn screening (NBS) without pathogenic variants in *GALT*, *GALE*, or *GALK1* or other explanation for increased galactose, were found to have pathogenic biallelic variants in *GALM* in combination with decreased GALM enzyme activity (measured in two patients) and decreased GALM protein expression (measured in three patients) [7]. Two of the patients developed bilateral cataracts. Two patients had increased total bile acids, and two patients had increased liver enzymes. No other symptoms were reported. Since this first publication [7], more than 40 patients with GALM deficiency have been reported (Table 1). All of the patients were detected by abnormal NBS, except for one pair of siblings who did not receive NBS and were diagnosed with GALM deficiency at 3-months and 6-years of age [9]. Here, we present a patient with GALM deficiency who had negative NBS in the Netherlands and was identified at age 1.5 years. As the majority of the previously reported patients had abnormal NBS, we sought to compare our case to previously reported patients and include a review of all GALM cases described in the literature. Informed consent was obtained from the patient’s parents to publish this paper.

## 2. Case Description

The patient is a 15-month-old girl who presented with global developmental delay and mild nystagmus. Her parents were first cousins. Her NBS, which includes screening for GALT and GALK1 deficiency in the Netherlands, was performed on day 4 of life and was unremarkable. GALT activity was 12.6 U/dl blood, with the NBS cut-off <2 U/dl; total galactose in our patient was 1000 µmol/L, with the upper limit of normal (ULN) for GALT deficiency being 1350 µmol/L and ULN for GALK1 deficiency 2100 µmol/L (if combined with normal GALT activity) (Table 1). Her medical history was remarkable for feeding problems and jaundice in the neonatal period. At 15 months, because of delayed achievement of motor milestones and language development, nystagmus, and a history of jaundice, broad metabolic screening was performed, in which strongly increased levels of galactose (13,167 mmol/mol creatinine, ULN 326) and galactitol (427 mmol/mol creatinine, ULN 71) were found in urine. On repeated measurement, one month later, urine galactose and galactitol levels increased even further (Table 2). GALT activity measurement was repeated in our laboratory and was normal. She was started on a lactose-restricted diet. Her NBS samples were re-evaluated and repeated. Total galactose values on repeat were again below the cut-off values for the NBS for GALT and GALK deficiency (Table 1). Whole exome sequencing was performed and showed homozygous variants in *GALM* (c.424G>A p.(Gly142Arg)), inherited from both parents. In addition, a de novo heterozygous frameshift variant in *ANKS1B* was found, which is associated with neurodevelopmental disorder [13]. The *ANKS1B* variant is thought to be the cause of her developmental delay, although we cannot completely rule out any effects of GALM deficiency on her development. After 2 months on the diet, galactose levels in urine normalized and galactitol levels were only slightly elevated (Table 2). She was evaluated for cataract, which was not observed, and her nystagmus had improved. She made progress in reaching developmental milestones.

## 3. Discussion

Here, we report on a patient with GALM deficiency who had negative newborn screening for galactosemia in the Netherlands. As all the GALM patients described in the first publication of GALM deficiency had abnormal NBS for galactosemia, we sought to compare our patient to previously reported patients. To do so, we performed a literature review of all reported patients with GALM deficiency (n = 47, including our patient). Demographic, genetic, and biochemical characteristics, results of NBS, and any reported diets are presented in Table 1 and Table 3.

Our patient had homozygous missense variants in *GALM*. The majority of patients hah compound heterozygous *GALM* variants (n = 25); homozygous *GALM* variants were also common (n = 20), while chromosomal abnormalities resulting in *GALM* deletion were only reported in two patients (Table 3). The common variants are shown in Figure 2.

Of all the reported GALM-deficient patients described so far, only one sibling pair was not identified by NBS (P4 and P5; Table 1 and Table 2) [9]; however, these siblings did not participate in NBS. In addition, one pair of monozygotic twins tested positive in the second NBS sample, taken at 15 days of life but not in the first sample taken within 48–72 h of life (P2 and P3; Table 1) [10].

At first, we were surprised that our GALM-deficient patient had a negative newborn screening for galactosemia. We considered that this was due to severe feeding problems of unknown cause in the neonatal period and that she had not yet taken sufficient lactose. (Screening too early in life or newborns on a galactose-restricted diet such as parenteral feeding or hypoallergenic formula may cause false negative NBS results, a problem the current classical galactosemia screening is also faced with.)

In order to correctly interpret the NBS results, we have also looked into the differences between NBS programs (Table 4). NBS programs are specifically developed for target disorders and differ between different countries, and in some cases, within the country regions. To correctly interpret (galactosemia) NBS results, it is essential to understand which disorders are the target of a certain NBS program and which metabolites/enzymes are screened. Therefore, each NBS result should be interpreted based on the method, reference values, age at sampling, and laboratory standards pertaining to its NBS program. In the Netherlands, total galactose (galactose plus Gal-1-P) is measured together with GALT activity with the goal to detect GALT and GALK1 deficiency. On the other hand, in Japan, galactose, Gal-1-P, total galactose, and/or GALT activity are measured, depending on the discretion of each jurisdiction [11]. In addition, differences in cut-off total galactose levels exist [7,10,11,12] (Table 4) and NBS results cannot be directly compared. In addition, different NBS programs use different concentration units, which further complicates the comparisons of NBS results (Table 1 and Table 4).

The majority of the reported patients were identified by the Japanese NBS and had mildly elevated galactose levels in NBS (5.2–19.4 mg/dL; cut-off values 3–10 mg/dL). Total galactose in these patients was not frequently assessed, though all the reported patients from the Japanese NBS had abnormal total galactose values (Table 1). In comparison, our patient had normal total galactose based on the Dutch NBS program (Table 1). Unfortunately, given the differences in the NBS programs as stated above, it is not possible to directly conclude whether our patient would have been captured by Japanese (or other) NBS programs. However, the design of the Dutch NBS program may lead to overlooking patients with GALM deficiency and could explain the negative NBS of our patient [14].

In addition to the NBS results, we also assessed the impact of a lactose-free diet on GALM patients. Our patient was started on a galactose-free diet at the time of the (genetic) diagnosis and showed almost complete biochemical normalization and (perhaps coincidentally) improvement of nystagmus. From the literature, all but one reported patient (P5; Table 1) followed a galactose-restricted diet [9]. Twelve patients (temporarily) stopped the diet (Table 1). One patient (P21; Table 1) is not currently on a diet, but diet history is unknown [11]. Of the two patients who were, as far as we know, never on a diet, one (P5; Table 1) presented with hypermetropia without any other symptoms (age at last follow-up, 10 years old) [9] and the other (P21; Table 1), of whom data are missing, experienced liver dysfunction and cholestasis (age at last follow-up, 19 years old) [11]. For most patients (n = 26), no symptoms were reported. However, these patients started a galactose-restricted diet in the first months of life. Seven of these patients stopped the diet between the ages of 12 and 31 months (ages at last follow-up, 4–20 years old) due to biochemical improvement.

Reported clinical symptoms included (see Table 1, N = 47) liver dysfunction/elevated AST, ALT/ hepatomegaly (n = 10), cataract (n = 5), cholestasis/neonatal jaundice (n = 4), portosystemic shunt (n = 3), and increased total bile acids (TBAs) (n = 3). One patient was small for gestational age and experienced transient tachypnea as a neonate and febrile status epilepticus twice [10]. Developmental disorder (as in our case) was reported in only one other patient [7,11]. Out of the five patients that developed cataract, cataract resolved on a galactose-restricted diet in two patients. Of the three patients with unresolved cataract, one did not receive NBS and started a diet at 106 days old [9], while the others started a galactose-restricted diet at 3 weeks [7,11] and two months [11] of age. Liver-related symptoms were mild and improved over time [11].

It is not yet known to which extend a lactose-restricted diet is necessary and able to prevent the onset of symptoms in GALM deficiency. Based on the available data, a diet might prevent cataract in some patients, but symptoms of undiagnosed and untreated GALM-deficient patients are largely unknown. Although the true prevalence of GALM deficiency in the Netherlands is unknown (estimated prevalence of non-Finnish European population is 1:1,716,145 [15]), likely more patients (will) exist. If more information on natural history and effect of diet in GALM deficiency becomes available and a lack of lactose-restricted diet indeed allows symptoms to arise, NBS design for screening galactosemia may be reconsidered. To target GALM deficiency, separate galactose and Gal-1-P values with clear cut-off values could be established. Other possibilities are lowering cut-off values for total galactose followed by genetic screening and/or measuring GALM activity in dried blood spots. A similar approach is currently used in Taiwan [12].

In conclusion, here, we present a first case of GALM deficiency in the Netherlands, and highlight how different designs for galactosemia screening may lead to overlooking patients with GALM deficiency.

## Figures and Tables

**Figure 1 IJNS-11-00025-f001:**
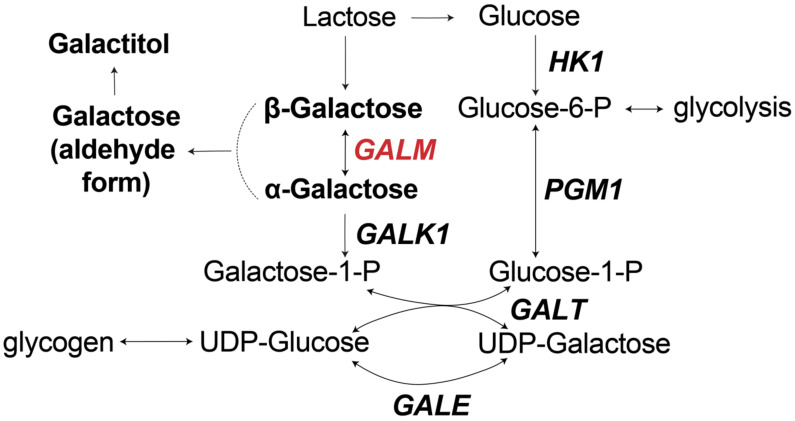
Schematic representation of lactose metabolism and Leloir pathway in human. GALE—UDP-glucose 4-epimerase; GALK1—galactose-kinase; GALM—galactose mutatorase; GALT—galactose-1-phosphate uridyltransferase; HK1—hexokinase 1; PGM1—phosphoglucomutase 1.

**Figure 2 IJNS-11-00025-f002:**
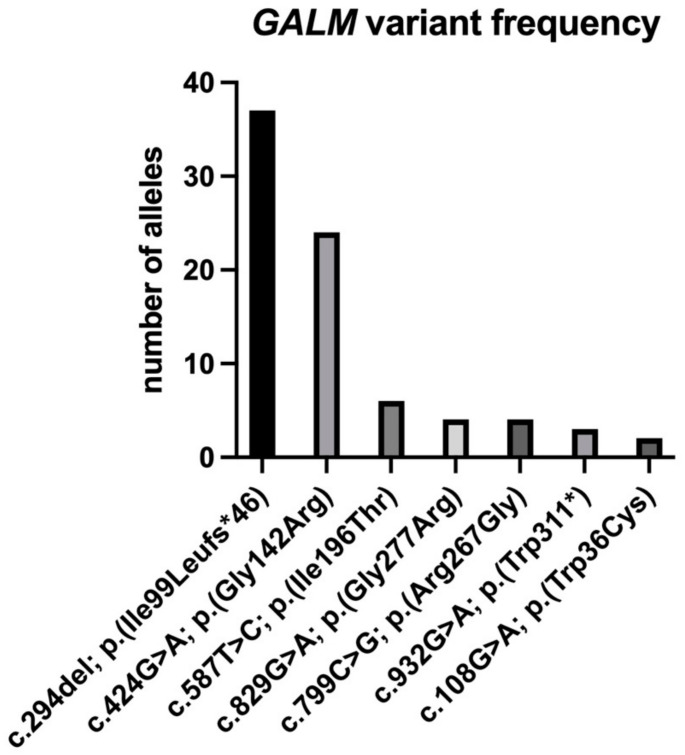
The allele frequency of the most commonly reported *GALM* variants. The most common reported variant was c.294del, p.(Ile99Leufs*46) (n = 37), followed by c.424G>A, p.(Gly142Arg). All the variants are listed in Table 3.

**Table 1 IJNS-11-00025-t001:** Biochemical and clinical findings of reported GALM patients. P2 and P3 were born and had a second NBS at 15-days of life, which is performed in special circumstances in the community of Galicia, where the twins were born. Abbreviations: NBS—newborn screening; tGal—total galactose; gal—galactose; N—normal reference range; NR—not reported; do—days old; mo—months old; wo—weeks old; yo—years old; ALT—alanine transaminase; AST—aspartate transaminase; TBA—total bile acid.

Patient	Publication	NBS Total Galactose (Galactose + gal-1-p) in mg/dL *	NBS Galactose in mg/dL	NBS Gal-1-P in mg/dL ^#^	Highest (Total) Galactose in Blood in mg/dL * (Age)	Gal-Restricted Diet (Age Started-Ended)	Clinical Presentation
P1	This publication	Repeated measurements 842; 932; 1000 µmol/L * N< 1350 µmol/L *	x	x	1000 µmol/L * (4 do, tgal)	17 mo—ongoing	Nystagmus, developmental delay (probably caused by a second diagnosis of *ANKS1B* frameshift variant), neonatal jaundice.
P2	Sánchez-Pintos et al., 2024 (pt1) [10]	1st sample: 11 2nd sample: 21.6 N < 18	x	1st sample: 0.38 mmol/L ^#^ 2nd sample: 0.12 mmol/L ^#^ N < 0.7	21.6 (15 do, tgal)	3–6 mo and 7–18 mo	No symptoms reported.
P3	Sánchez-Pintos et al., 2024 (pt2) [10]	1st sample: 18.9 2nd sample: 23.7 N < 18	x	1st sample 0.33 mmol/L ^#^ 2nd sample: 0.15 mmol/L ^#^ n < 0.7	23.7 (15 do, tgal)	3–6 mo and 7–18 mo	Small for gestational age; neonatal transient tachypnea requiring respiration support; 2× febrile status epilepticus.
P4	Yazici et al., 2021 (pt1) [9]	No NBS performed	No NBS performed	No NBS performed	33.7 (3 mo, tgal) N < 10 32.2 (3 mo, gal) N < 5	106 do—ongoing	Mild bilateral cataracts, mildly elevated AST, ALT. Cataracts did not resolve on galactose-free diet.
P5	Yazici et al., 2021 (pt2) [9]	No NBS performed	No NBS performed	No NBS performed	under non-restricted diet: 3.1 (6 yo, tgal) N < 10 1.9 (6 yo, gal) N < 5	no	Hypermetropia
P6	Wada et al., 2019 (pt1) [7] and Mikami-Saito (pt3) [11]	NR	12.4 N < 3–6 depending on the prefecture	8.9 N < 10–15 depending on the prefecture	17.3 (44 do, gal)	44 do- +/− 110 do and +/− 250 do—ongoing	Cataract at 7 mo during temporary suspension of diet, resolved at 23 mo.
P7	Wada et al., 2019 (pt2) [7] and Mikami-Saito (pt15) [11]	NR	8.7 N < 3–6 depending on the prefecture	8.9 N < 10–15 depending on the prefecture	41.9 (4 mo, gal)	4 mo—ongoing	No symptoms reported
P8	Wada et al., 2019 (pt3) [7]	NR	11.9 N < 3–6 depending on the prefecture	6.6 N < 10–15 depending on the prefecture	19.2 (16 do, gal)	1 mo—ongoing	Transiently elevated ALT, AST, TBA.
P9	Wada et al., 2019 (pt4) [7] and Mikami-Saito (pt18) [11]	NR	9.7 N < 3–6 depending on the prefecture	2.4 N < 10–15 depending on the prefecture	28.2 (5 mo, gal)	1–20 mo	No symptoms reported
P10	Wada et al., 2019 (pt5) [7] and Mikami-Saito (pt20) [11]	NR	10.0 N < 3–6 depending on the prefecture	10.8 N < 10–15 depending on the prefecture	during transient dietary relaxation: 29.8 (8 mo, gal)	1.9 mo—ongoing	Neonatal jaundice, portosystemic shunt.
P11	Wada et al., 2019 (pt60) [7] and Mikami-Saito (pt14) [11]	NR	12.8 N < 3–6 depending on the prefecture	0.3 N < 10–15 depending on the prefecture	34 (12 do, gal)	0.6/2 mo—ongoing	No symptoms reported.
P12	Wada et al., 2019 (pt7) [7] and Mikami-Saito (pt10) [11]	NR	15.7 N < 3–6 depending on the prefecture	10.4 N < 10–15 depending on the prefecture	during transient dietary relaxation: 31 (6 mo, gal)	1 mo- 11 yo after that relaxation with no high intake of dairy products	Transiently elevated TBA levels on non-restricted diet, developmental disorder.
P13	Wada et al., 2019 (pt8) [7] and Mikami-Saito (pt4) [11]	NR	11.4 N < 3–6 depending on the prefecture	6.7 N < 10–15 depending on the prefecture	34.1 (13 do, gal)	3 wo—ongoing	Sustained high levels of TBA and abnormal flow signal on the abdominal ultrasound until 8 mo, but no shunt detected; mild bilateral cataracts at 10 mo.
P14	Chen et al., 2024 [12]	1st sample: 17.57 μmol/L * 2nd sample: 34.1 μmol/L N < 30 μmol/L * or repeated sample < 15 μmol/L* (this differs from the Dutch reference range)	NR	NR	34.1 μmol/L * (9 do, tgal) (this differs from the Dutch reference range)	15 do—ongoing	No symptoms reported.
P15	Mikami-Saito et al., 2024 (pt1) [11]	NR	8.2 N < 3–8, depending on prefecture	7.1 N < 10–25, depending on prefecture	18 (at onset, gal)	1 mo—ongoing	Cataract at 1 mo, resolved at 8 months.
P16	Mikami-Saito et al., 2024 (pt2) [11]	NR	12.2 N < 3–8, depending on prefecture	7.5 N < 10–25, depending on prefecture	34.8 (at onset, gal)	2 mo—ongoing	Cataract at 2 mo, liver dysfunction.
P17	Mikami-Saito et al., 2024 (pt5) [11]	NR	10 N < 3–8, depending on prefecture	NR	NR	1 mo—ongoing	Liver dysfunction.
P18	Mikami-Saito et al., 2024 (pt6) [11]	NR	19.3 N < 3–8, depending on prefecture	8.8 N < 10–25, depending on prefecture	NR	2 mo—ongoing	Portosystemic shunt.
P19	Mikami-Saito et al., 2024 (pt7) [11]	NR	7 N < 3–8, depending on prefecture	1.1 N < 10–25, depending on prefecture	NR	Unknown—2 yo	No symptoms reported.
P20	Mikami-Saito et al., 2024 (pt8) [11]	NR	6.2 N < 3–8, depending on prefecture	14.8 N < 10–25, depending on prefecture	NR	0.4 mo—ongoing	Hepatomegaly, liver dysfunction.
P21	Mikami-Saito et al., 2024 (pt9) [11]	14.6 N < 6–12, depending on prefecture	6.6 N < 3–8, depending on prefecture	NR	NR	Unknown, no diet currently	Liver dysfunction, cholestasis.
P22	Mikami-Saito et al., 2024 (pt11) [11]	18.6 N < 6–12, depending on prefecture	16.8 N < 3–8, depending on prefecture	2.5 N < 10–25, depending on prefecture	NR	0.1 mo—ongoing	No symptoms reported.
P23	Mikami-Saito et al., 2024 (pt12) [11]	NR	10.2 N < 3–8, depending on prefecture	7.2 N < 10–25, depending on prefecture	NR	0.7–31 mo	No symptoms reported.
P24	Mikami-Saito et al., 2024 (pt13) [11]	NR	7.8 N < 3-8, depending on prefecture	5.9 N < 10-25, depending on prefecture	NR	1–12 mo	No symptoms reported.
P25	Mikami-Saito et al., 2024 (pt16) [11]	NR	15.4 N < 3–8, depending on prefecture	4.7 N < 10–25, depending on prefecture	NR	1–12 mo	No symptoms reported.
P26	Mikami-Saito et al., 2024 (pt17) [11]	23.88 N < 6–12, depending on prefecture	19.05 N < 3–8, depending on prefecture	6.96 N < 10–25, depending on prefecture	NR	0.1 mo—ongoing	Cholestasis.
P27	Mikami-Saito et al., 2024 (pt19) [11]	16.4 N < 6–12, depending on prefecture	12.2 N < 3–8, depending on prefecture	6 mg/dL N < 10–25, depending on prefecture	NR	Unknown—currently on diet	No symptoms reported.
P28	Mikami-Saito et al., 2024 (pt21) [11]	12.4 N < 6–12, depending on prefecture	8.4 N < 3–8, depending on prefecture	8.7 N < 10–25, depending on prefecture	NR	3–7 mo	Liver dysfunction.
P29	Mikami-Saito et al., 2024 (pt22) [11]	NR	11.8 N < 3–8, depending on prefecture	6.6 N < 10–25, depending on prefecture	NR	0.5 mo—ongoing	Liver dysfunction.
P30	Mikami-Saito et al., 2024 (pt23) [11]	NR	9.9 N < 3–8, depending on prefecture	9.3 N < 10–25, depending on prefecture	NR	0.7 mo—ongoing	No symptoms reported.
P31	Mikami-Saito et al., 2024 (pt24) [11]	13.58 N < 6–12, depending on prefecture	10.9 N < 3–8, depending on prefecture	NR	NR	0.5 mo—ongoing	No symptoms reported.
P32	Mikami-Saito et al., 2024 (pt25) [11]	NR	12 N < 3–8, depending on prefecture	4.8 N < 10–25, depending on prefecture	NR	0.3 mo—ongoing	No symptoms reported.
P33	Mikami-Saito et al., 2024 (pt26) [11]	NR	11.5 N < 3–8, depending on prefecture	2.4 N < 10–25, depending on prefecture	NR	1 mo—ongoing	No symptoms reported.
P34	Mikami-Saito et al., 2024 (pt27) [11]	NR	19.4 N < 3–8, depending on prefecture	4.2 N < 10–25, depending on prefecture	NR	1.3 mo—ongoing	No symptoms reported.
P35	Mikami-Saito et al., 2024 (pt28) [11]	NR	12.7 N < 3–8, depending on prefecture	0.9 N < 10–25, depending on prefecture	NR	0.1 mo—ongoing	No symptoms reported.
P36	Mikami-Saito et al., 2024 (pt29) [11]	NR	16.22 N < 3–8, depending on prefecture	4.1 N < 10–25, depending on prefecture	NR	0.2 mo—ongoing	No symptoms reported.
P37	Mikami-Saito et al., 2024 (pt30) [11]	NR	NR	NR	NR	3 mo—ongoing	No symptoms reported.
P38	Mikami-Saito et al., 2024 (pt31) [11]	18.5 N < 6–12, depending on prefecture	10.7 N < 3–8, depending on prefecture	11.3 N < 10–25, depending on prefecture	NR	1 mo—ongoing	No symptoms reported.
P39	Mikami-Saito et al., 2024 (pt32) [11]	15.39 N < 6–12, depending on prefecture	9.78 N < 3–8, depending on prefecture	8.08 N < 10–25, depending on prefecture	NR	4–23 mo	No symptoms reported.
P40	Mikami-Saito et al., 2024 (pt33) [11]	17.9 N < 6–12, depending on prefecture	11.9 N < 3–8, depending on prefecture	8.6 N < 10–25, depending on prefecture	NR	0.1 mo—ongoing	No symptoms reported.
P41	Mikami-Saito et al., 2024 (pt34) [11]	22.9 N < 6–12, depending on prefecture	5.2 N < 3–8, depending on prefecture	19.3 N < 10–25, depending on prefecture	NR	0.1 mo—ongoing	No symptoms reported.
P42	Mikami-Saito et al., 2024 (pt35) [11]	NR	12.9 N < 3–8, depending on prefecture	2.7 N < 10–25, depending on prefecture	NR	1.6 mo—ongoing	No symptoms reported.
P43	Mikami-Saito et al., 2024 (pt36) [11]	18.1 N < 6–12, depending on prefecture	17.6 N < 3–8, depending on prefecture	0.8 N < 10–25, depending on prefecture	NR	0.3 mo—ongoing	No symptoms reported.
P44	Mikami-Saito et al., 2024 (pt37) [11]	15.4 N < 6–12, depending on prefecture	11.5 N < 3–8, depending on prefecture	5.6 N < 10–25, depending on prefecture	NR	0.7–22 mo	Portosystemic shunt.
P45	Mikami-Saito et al., 2024 (pt38) [11]	NR	10.8 N < 3–8, depending on prefecture	7.9 N < 10–25, depending on prefecture	NR	4 mo—ongoing	Liver dysfunction.
P46	Mikami-Saito et al., 2024 (pt39) [11]	NR	7.2 N < 3–8, depending on prefecture	6.2 N < 10–25, depending on prefecture	NR	1.6 mo—ongoing	Liver dysfunction.
P47	Mikami-Saito et al., 2024 (pt40) [11]	NR	15.34 N < 3–8, depending on prefecture	6.36 N < 10–25, depending on prefecture	NR	1 mo—ongoing	No symptoms reported.

Total bile acids; x—not tested. All reported patients had normal GALT enzymes. Samples with values outside reference ranges are underlined. * NBS in Netherlands and Taiwan use µmol/L. ^#^ NBS in Galicia, Spain uses mmol/L. (Direct comparison between different NBS systems using different units is not possible because in most programs, total galactose is defined as sum of free galactose + Gal-1-P and molecular weight of galactose differs from Gal-1-P with Gal-1-P to free galactose ratio that varies per patient.)

**Table 2 IJNS-11-00025-t002:** Galactose measurements in our patient at 15-, 16-, and 18-months of age (* two months after initiating galactose-restricted diet). ULN—upper limit of normal.

Age of the Patient	Urinary Galactose (mmol/mol Creatinine) (ULN 326)	Urinary Galactitol (mmol/mol Creatinine) (ULN 71)
15-months	13,167	427
16-months	26,078	929
18-months *	56	126

**Table 3 IJNS-11-00025-t003:** Genetic information of reported GALM patients. Abbreviations: mo—months old; yo—years old.

Patient	Publication	Age/Sex at Last Follow-Up	Genetic Variant 1	Genetic Variant 2
P1	This publication	18 mo/F	c.424G>A; p.(Gly142Arg)	c.424G>A, p.(Gly142Arg)
P2	Sanches Pintos et al., 2024 (pt1) [10]	9 yo/M	arr[hg19] 2p22.1(38,893,070 − 38,925,887)	4arr[hg19] 2p22.1(38,916,650−38,925,887)del
P3	Sanches Pintos et al., 2024 (pt2) [10]	9 yo/M	arr[hg19] 2p22.1(38,893,070 − 38,925,887)	4arr[hg19] 2p22.1(38,916,650−38,925,887)del
P4	Yazici et al., 2021 (pt1) [9]	3 yo/F	c.829G>A p.(Gly277Arg)	c. 829G>A, p.(Gly277Arg)
P5	Yazici et al., 2021 (pt2) [9]	10 yo/M	c.829G>A, p.(Gly277Arg)	c.829G>A, p.(Gly277Arg)
P6	Wada et al., 2019 (pt1) [7] and Mikami-Saito et al., 2024 (pt3) [11]	7 yo/M	c.244C>T p.(Arg82*)	c.294del, p.(Ile99Leufs*46)
P7	Wada et al., 2019 (pt2) [7] and Mikami-Saito et al., 2024 (pt15) [11]	10 yo/F	c.294del, p.(Ile99Leufs*46)	c.799C>G p.(Arg267Gly)
P8	Wada et al., 2019 (pt3) [7]	1 yo/M	c.294del, p.(Ile99Leufs*46)	c.294del, p.(Ile99Leufs*46)
P9	Wada et al., 2019 (pt4) [7] and Mikami-Saito (pt18) [11]	8 yo/F	c.932G>A, p.(Trp311*)	c.932G>A, p.(Trp311*)
P10	Wada et al., 2019 (pt5) [7] and Mikami-Saito (pt20) [11]	7 yo/M	c.424G>A, p.(Gly142Arg)	c.424G>A, p.(Gly142Arg)
P11	Wada et al., 2019 (pt6) [7] and Mikami-Saito (pt14) [11]	10 yo/F	c.424G>A, p.(Gly142Arg)	c.424G>A, p.(Gly142Arg)
P12	Wada et al., 2019 (pt7) [7] and Mikami-Saito (pt10) [11]	18 yo/M	c.424G>A, p.(Gly142Arg)	c.799C>G, p.(Arg267Gly)
P13	Wada et al., 2019 (pt8) [7] and Mikami-Saito (pt4) [11]	6 yo/M	c.424G>A, p.(Gly142Arg)	c.424G>A, p.(Gly142Arg)
P14	Chen et al., 2024 [12]	3 yo/M	c.325G>A, p.(Gly109Arg)	c.587T>C, p.(Ile196Thr)
P15	Mikami-Saito et al., 2024 (pt1) [11]	1 yo/M	c.256G>A, p.(Gly86Arg)	c.424G>A, p.(Gly142Arg)
P16	Mikami-Saito et al., 2024 (pt2) [11]	0 yo/M	c. 424G>A, p.(Gly142Arg)	c.587T>C, p.(Ile196Thr)
P17	Mikami-Saito et al., 2024 (pt5) [11]	34 yo/M	c.294del, p.(Ile99Leufs*46)	c.424G>A, p.(Gly142Arg)
P18	Mikami-Saito et al., 2024 (pt6) [11]	21 yo/F	c.294del, p.(Ile99Leufs*46)	c.294del, p.(Ile99Leufs*46)
P19	Mikami-Saito et al., 2024 (pt7) [11]	20 yo/F	c.294del, p.(Ile99Leufs*46)	c.294del, p.(Ile99Leufs*46)
P20	Mikami-Saito et al., 2024 (pt8) [11]	19 yo/M	c.424G>A, p.(Gly142Arg)	c.424G>A, p.(Gly142Arg)
P21	Mikami-Saito et al., 2024 (pt9) [11]	19 yo/F	c.294del, p.(Ile99Leufs*46)	c.294del, p.(Ile99Leufs*46)
P22	Mikami-Saito et al., 2024 (pt11) [11]	14 yo/F	c.108G>A, p.(Trp36Cys)	c.294del, p.(Ile99Leufs*46)
P23	Mikami-Saito et al., 2024 (pt12) [11]	13 yo/M	c.294del, p.(Ile99Leufs*46)	c.799C>G, p.(Arg267Gly)
P24	Mikami-Saito et al., 2024 (pt13) [11]	11 yo/F	c.294del, p.(Ile99Leufs*46)	c.294del, p.(Ile99Leufs*46)
P25	Mikami-Saito et al., 2024 (pt16) [11]	9 yo/M	c.294del, p.(Ile99Leufs*46)	c.294del, p.(Ile99Leufs*46)
P26	Mikami-Saito et al., 2024 (pt17) [11]	9 yo/M	c.108G>T, p.(Trp36*)	c.424G>A, p.(Gly142Arg)
P27	Mikami-Saito et al., 2024 (pt19) [11]	7 yo/M	c.108G>A, p.(Trp36Cys)	c.294del, p.(Ile99Leufs*46)
P28	Mikami-Saito et al., 2024 (pt21) [11]	7 yo/F	c.294del, p.(Ile99Leufs*46)	c.424G>A, p.(Gly142Arg)
P29	Mikami-Saito et al., 2024 (pt22) [11]	6 yo/M	c.294del, p.(Ile99Leufs*46)	c.294del, p.(Ile99Leufs*46)
P30	Mikami-Saito et al., 2024 (pt23) [11]	5 yo/M	c.294del, p.(Ile99Leufs*46)	c.424G>A, p.(Gly142Arg)
P31	Mikami-Saito et al., 2024 (pt24) [11]	5 yo/M	c.294del, p.(Ile99Leufs*46)	c.294del, p.(Ile99Leufs*46)
P32	Mikami-Saito et al., 2024 (pt25) [11]	5 yo/M	c.294del, p.(Ile99Leufs*46)	c.799C>G, p.(Arg267Gly)
P33	Mikami-Saito et al., 2024 (pt26) [11]	4 yo/F	c.294del, p.(Ile99Leufs*46)	c.878A>C p.(Lys293Thr)
P34	Mikami-Saito et al., 2024 (pt27) [11]	4 yo/M	c.294del, p.(Ile99Leufs*46)	c.294del, p.(Ile99Leufs*46)
P35	Mikami-Saito et al., 2024 (pt28) [11]	4 yo/M	c.294del, p.(Ile99Leufs*46)	c.365_392del, p.(Val122Alafs*14)
P36	Mikami-Saito et al., 2024 (pt29) [11]	4 yo/M	c.424G>A, p.(Gly142Arg)	c.587T>C, p.(Ile196Thr)
P37	Mikami-Saito et al., 2024 (pt30) [11]	4 yo/F	c.424G>A, p.(Gly142Arg)	c.845C>A p.(Thr282Lys)
P38	Mikami-Saito et al., 2024 (pt31) [11]	4 yo/F	c.294del, p.(Ile99Leufs*46)	c.424G>A, p.(Gly142Arg)
P39	Mikami-Saito et al., 2024 (pt32) [11]	4 yo/F	c.424G>A, p.(Gly142Arg)	c.587T>C, p.(Ile196Thr)
P40	Mikami-Saito et al., 2024 (pt33) [11]	3 yo/F	c.294del, p.(Ile99Leufs*46)	c.294del, p.(Ile99Leufs*46)
P41	Mikami-Saito et al., 2024 (pt34) [11]	3 yo/F	c.424G>A, p.(Gly142Arg)	c.424G>A, p.(Gly142Arg)
P42	Mikami-Saito et al., 2024 (pt35) [11]	3 yo/M	c.294del, p.(Ile99Leufs*46)	c.587T>C, p.(Ile196Thr)
P43	Mikami-Saito et al., 2024 (pt36) [11]	3 yo/M	c.587T>C, p.(Ile196Thr)	c.932G>A, p.(Trp311*)
P44	Mikami-Saito et al., 2024 (pt37) [11]	3 yo/M	c.294del, p.(Ile99Leufs*46)	c.294del, p.(Ile99Leufs*46)
P45	Mikami-Saito et al., 2024 (pt38) [11]	3 yo/F	c.221C>A, p.(Ala74Glu)	c.424G>A, p.(Gly142Arg)
P46	Mikami-Saito et al., 2024 (pt39) [11]	2 yo/F	c.294del, p.(Ile99Leufs*46)	c.424G>A, p.(Gly142Arg)
P47	Mikami-Saito et al., 2024 (pt40) [11]	1 yo/M	c.234G>T, p.(Arg78Ser)	c.294del, p.(Ile99Leufs*46)

**Table 4 IJNS-11-00025-t004:** Galactose NBS systems of patients with GALM deficiency described in the literature. Abbreviations: ho—hours old; DBS—dried blood spot.

Country	Target Conditions (Effected Gene)	Target (Biochemical)	Age at Screening	GALT Enzyme Activity Cut-Off	Total Galactose Cut-Off	Galactose Cut-Off	Galactose-1-Phosphate Cut-Off	Comments
The Netherlands	*GALT*	Total galactose (galactose + gal-1-p), GALT	72–168 ho	<2.0 U/dL DBS	>1350 μmol/L DBS	X	X	Positive if both markers are outside reference range
	*GALK*	Total galactose (galactose + gal-1-p)		>2.0 U/dL DBS	>2100 μmol/L DBS	X	X	Positive if both markers are outside reference range
Galicia, Spain	*GALT*	Galactose, gal-1-P (separately)	48–72 h of life, second sample 15 days in special cases [11]	X	>18 mg/dL DBS	Qualitative (urine)	>0.7 mmol/L DBS	Second samples are taken in some circumstances
Japan	*GALT*, *GALK*	Galactose, gal-1-p, (total galactose), GALT enzyme	96–144 ho [12]	Sometimes measured depending on prefecture	>6–12 mg/dL DBS	>3–8 mg/dL DBS	>10–25 mg/dL DBS	Different cut-offs are used by different prefectures (country regions)
Taiwan	*GALT*, *GALK*, *GALE*, (*GALM*)	Total galactose, referral and genetic testing for *GALT*, *GALK*, *GALE*, (*GALM*)	48–72 ho [13]	2nd tier measurement	>30 μmol/L or repeated sample > 15–18 μmol/L DBS	X	X	Total galactose is the primary marker

## Data Availability

All data is available upon request.

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
