# Peer review of "Rethinking Newborn Screening: A Case of GALM Deficiency"

_2409-515X, 2025, doi:10.3390/ijns11020025_

Round 1

Reviewer 1 Report

Comments and Suggestions for Authors

The authors raised an interesting issue with the NBS design, including the cutoffs, and how this could affect the ability of NBS to identify patients with GALM deficiency. However, the use of two different unit systems complicates direct comparisons.

  1. The data in Table 1 can be better presented:
    1. Covert NBS total galactose to the same unit (umol/L or mg/dL) such it's easier to directly compare cutoffs and patient results.
    2. Provide normal ranges for Column 6 (highest total Gal), if such information is not provided in Column 3
    3. Highlight samples with abnormal values
    4. Any other patients (other than P1) have other genetic conditions and/or comorbidity? If so, please list them accordingly. 
  2. Line 122 – again it would be helpful to convert unit and evaluate the proband’s total galactose level against various cutoffs.
  3. It can be beneficial to list different NBS program's target conditions and their respective cutoff. Will this patient be captured on NBS if a different cutoff is adopted? The authors also provided an estimated prevalence of GALM and noted more patient may or will exist. 
  4. Do you have RBC or whole blood Gal-1-P level on the proband and from literature? Can it be integrated with the rest of the biochemical data to guide the differential diagnosis? How about data on GALK and GALE enzyme activity?
  5. Line 165-166 – What would be a potential solution for NBS of GALM? I image the cutoff can be lowered, but this would potentially increase the false positive rate.
  6. Table 3 - list other genetic finding if applicable. Also please provide a breakdown of the frequencies of the common alleles associated with GALM. 

Author Response

Please see the attachment (Reply to both reviewers). 

Reviewer 2 Report

Comments and Suggestions for Authors

The authors describe a single patient with GALM deficiency. This patient had a negative newborn screening (NBS), and galactosemia was incidentally identified during diagnostic investigations for developmental delay, which was later attributed to another de novo variant. Whole-exome sequencing revealed a homozygous variant, p.Gly142Arg, inherited from consanguineous parents, identified in the patient. This variant is known to be prevalent and pathogenic in multiple populations. Although it has been anticipated that differences in NBS systems among various countries or regions could lead to variable detection outcomes for GALM deficiency, this is the first report describing an actual GALM-deficient patient who tested negative on NBS. I agree with the authors' assertion that existing NBS criteria, primarily designed to detect GALT and GALK deficiencies, may fail to identify patients with GALM deficiency.

While this manuscript does not provide new insights into the clinical manifestations of GALM deficiency or novel variants of the GALM gene, the circumstances leading to the patient's diagnosis, particularly the negative NBS result, might hold potential significance for publication. However, several issues within the manuscript still need to be addressed.

Minor

  1. L49: “pathogen” should be “pathogenic”

  1. Table 1: “NBS Gal-1-P in mmol/L” in the header line should be mg/dL for Japanese patients.

  1. Table 1: Do the "x" marks in P1, P2, and P3 indicate "not tested"?

Author Response

Please see the attachment (the reply to both reviewers is in the same document)

Round 2

Reviewer 1 Report

Comments and Suggestions for Authors

Total galactose is measured after hydrolyzing Gal1P to galactose and the galactose is measured. Both units both described the amount of total galactose regardless of whether it’s from gal1P or free galactose. Using the MW of galactose to convert the unit should be appropriate.